# Fusing Structural and Functional MRIs using Graph Convolutional Networks for Autism Classification

**Devanshu Arya**[*1]                                                           D.ARYA@UVA.NL

**Richard Olij**[*1]                                               OLIJ.RICHARD@GMAIL.COM

**Deepak K. Gupta**[1]                                              D.K.GUPTA@UVA.NL

**Ahmed El Gazzar**[2]                                         GAZZAR033@GMAIL.COM

**Guido van Wingen**[2]                                  GUIDOVANWINGEN@GMAIL.COM

**Marcel Worring**[1]                                             M.WORRING@UVA.NL

**Rajat Mani Thomas**[2,3]                                    RAJATTHOMAS@GMAIL.COM

[1] *Informatics Institute, University of Amsterdam, The Netherlands*
[2] *Department of Psychiatry, Amsterdam University Medical Center, The Netherlands*
[3] *Spectrum AI, Bengaluru, India*

## Abstract

Geometric deep learning methods such as graph convolutional networks have recently proven to deliver generalized solutions in disease prediction using medical imaging. In this paper, we focus particularly on their use in autism classification. Most of the recent methods use graphs to leverage phenotypic information about subjects (patients or healthy controls) as additional contextual information. To do so, metadata such as age, gender and acquisition sites are utilized to define intricate relations (edges) between the subjects. We alleviate the use of such non-imaging metadata and propose a fully imaging-based approach where information from structural and functional Magnetic Resonance Imaging (MRI) data are fused to construct the edges and nodes of the graph. To characterize each subject, we employ *brain summaries*. These are 3D images obtained from the 4D spatiotemporal resting-state fMRI data through summarization of the temporal activity of each voxel using neuroscientifically informed temporal measures such as amplitude low frequency fluctuations and entropy. Further, to extract features from these 3D brain summaries, we propose a 3D CNN model. We perform analysis on the open dataset for autism research (full ABIDE I-II) and show that by using simple brain summary measures and incorporating sMRI information, there is a noticeable increase in the generalizability and performance values of the framework as compared to state-of-the-art graph-based models

**Keywords:** Graph Convolutions, Neuroimaging, Autism Classification.

## 1. Introduction

Neuroimaging holds the promise of objective diagnosis and prognosis in psychiatry. However, unlike neurological disorders, psychiatric disorders do not show obvious alterations in

---

* Contributed equally

physical appearance of the brain. Thus, structural Magnetic Resonance Imaging (sMRI) scans of the brain do not reveal differences between a healthy and a pathological brain. Researchers have long posited that patterns which distinguish between the two brains are not in sMRI, but in resting-state MRI (rs-fMRI) scans instead (Zhan and Yu, 2015). These scans involve mapping the blood oxygenation level (a proxy for brain activity) throughout the brain at an interval of 1-2 seconds, resulting in a 4D spatio-temporal image. Typically, at a scanning resolution of 4 mm and 300 temporal sampling points, this results in a 20 million dimensional feature vector. Finding patterns in a high-dimensional space to distinguish between healthy and psychiatric subjects is a challenge that still needs to be resolved.

One of the major challenges in developing an objective schema for the diagnosis of autism is the scarcity of reliable, consistent and sufficiently large datasets. Some recent initiatives such as the Autism Brain Imaging Data Exchange (ABIDE) (Di Martino et al., 2014) have tried to aggregate brain imaging dataset of Autistic (ASD) and typically developing or control (TD or CON) participants from various sites around the world. The complete dataset including ABIDE-I and ABIDE-II comprises over 2100 subjects including ASD and CON. ABIDE has thus become a benchmark dataset for autism classification.

Several machine learning approaches have been used for autism classification, such as support vector machine (Jiao et al., 2010; Bi et al., 2018), decision tree (Jiao et al., 2010), random forest (Maenner et al., 2016), deep neural networks (Khosla et al., 2018; Gazzar et al., 2019; Kong et al., 2019), among others. All these methods rely solely on subject-specific imaging features that fail to encode the similarities or dissimilarities between subjects. However, relational information is highly desirable in autism classification because (a) datasets are relatively small for a deep learning model, and (b) the dataset is obtained from multiple sites leading to inconsistent data-points.

Recent approaches using Graph Convolutional networks (GCNs) (Parisot et al., 2017; Anirudh and Thiagarajan, 2019) have been shown to utilize the relations between subjects along with their brain activity patterns. GCN uses a population graph where subjects (defined as nodes) are connected to similar ones through edges. The prediction for any new subject can be made based on both the subject-specific data, as well as the relational information from other similar subjects. However, almost all recent studies limit their study to a subset of the subjects, which primarily involves rejecting subjects with data of too short duration as well as those containing significant noise in them (Moradi et al., 2017; Abraham et al., 2017; Parisot et al., 2017; Khosla et al., 2018). While this helps in better training of the models, the adverse affect includes reduced generality to noisy and complex test subjects. Another approach (Ktena et al., 2017) uses metric learning method to evaluate distance between graphs, where each graph represents a brain network of each subject and the dataset used is a curated list from ABIDE-I. In this paper, unlike (Ktena et al., 2017) we cast the problem as node classification on a population graph and develop Deep Learning (DL) models which can deliver comparable performances even when using the entire ABIDE dataset spanning across all sites containing heterogeneous samples.

Albeit the availability of the ABIDE dataset, the dimensionality of the input data is too large to use it without any preprocessing or feature engineering. Different approaches have been used in the past to reduce the dimensionality of the data. Since rs-fMRI data comprises spatio-temporal signal, dimensionality reduction can be performed in space, time or even both. An approach for spatial downscaling is to use brain atlases, where the about

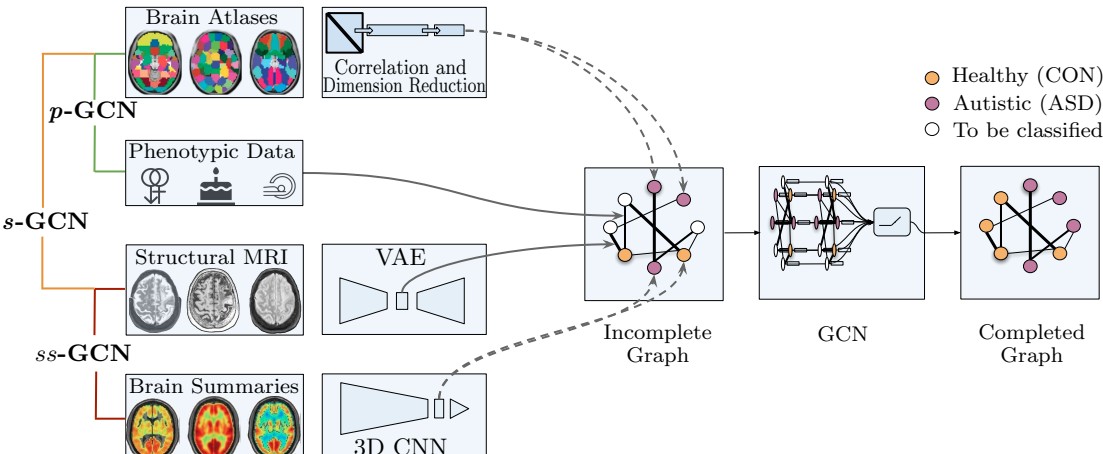

Figure 1: Schematic representation showing different models based on Graph Convolutional Networks (GCN) for the classification of subjects for autism disorders.

one million voxels in space are locally averaged to obtain around 100 to 400 non-overlapping regions. The reduced set of time courses thereafter can directly be treated using a 1D convolutional neural network (CNN) (Gazzar et al., 2019), or used to build a correlation matrix that with further processing provides an even reduced set of features (Parisot et al., 2018). Features obtained from the correlation matrix are for example reduced using recursive feature elimination (RFE) approach, where a subset of features are iteratively removed until a desired dimension is reached.

An alternate approach to treating the 4D brain volumes would be to preserve the full resolution, and only perform reductions in the temporal dimension. For example, the temporal signals could be summarized at voxel level using summary measures such as Amplitude Low Frequency Fluctuation (ALFF). ALFF is a measure that is posited to reveal differences in the underlying processing of the brain and is calculated based on the ratio of spectral power in two distinct frequency ranges. To the best of our knowledge, such summaries have not been incorporated in DL models for neuroimaging, and in this paper, we explore the applicability of such summaries. Moreover, we eliminate the use of reduction techniques such as RFE to avoid undesired excessive loss of information. Rather, we propose to use a Variational Autoencoder (VAE) to project the information on to a lower dimensional representation, and use it as a feature vector for our model.

In GCN based methods, while the features of the subjects are used to characterize the nodes, the definition of edges relies mostly on their phenotypic data (*e.g.* sex, age and acquistion sites). However, phenonotypic information are merely proxies and instead of using them to define connections among the subjects, we propose to use the 'actual similarities' between the brains' structures. In the past, sMRI data has been used to understand the variability of brain structure based on age (Brickman et al., 2007; Su et al., 2012), gender (Tyan et al., 2017) and acquisition sites (Littmann et al., 2006). This implies that these phenotypic parameters correlate with the structural imaging data with an association between sMRI and age/sex/site. These studies also indicate, for example, that "brain-age" need not always coincide with the age reported. In order to avoid such uncertainties in

establishing the edges on a graph, we resort to the use of sMRI images directly giving us one variable to establish the relationship. Hence, as opposed to defining relations based on arbitrary metadata to infer structural brain similarities, comparing actual structural data from subjects will yield a better approximation of similar brain structures. Therefore it can be assumed to have lower variance in the functional features, since the brain is expected to behave in a more similar way. Based on this motivation, we hypothesize that the structural images have higher expressibility of subject relations, and propose to use them to build the edges of the population graph. For better clarity, here and henceforth, we will refer the approaches of Parisot et al. (2018) and that based on structural MRI data as $p$-GCN and $s$-GCN, respectively. Furthermore, the approach involving structural MRI data as well as the brain summaries will be referred as $ss$-GCN.

In this paper, we address the various limitations of the existing methods as outlined above. To summarize, the main contributions[1] of our paper are:

- the fusion of structural and functional resting-state images for autism classification, thus alleviating the need to use non-imaging metadata of patients,

- the use of various temporal summary measures to reduce the 4D input volume to a 3D volume at the original spatial resolution for classification.

- Finally, we present a novel 3D CNN-GCN model for improved classification of subjects for autism disorders. The CNN module is used to encode the summarized 3D volumes into lower dimensional feature vectors for the nodes of the graph model.

## 2. Graph Convolution for Autism Classification

This section provides a brief overview on the application of GCNs for the classification of subjects for autism disorders. Figure 1 illustrates the 3 types of GCN based models: $p$-GCN and our proposed models $s$-GCN and $ss$-GCN. As can be seen, the pipeline involves an initial population graph which comprises two parts: 1) a feature vector that characterizes each node (subject) of the graph, and 2) a similarity measure to define edges (relations) between the nodes.

In $p$-GCN, the feature vector for each node is obtained by building a correlation matrix between the time series values from all possible pairs of regions in the respective brain atlas. The information contained in the upper triangle of the correlation matrix is extracted, flattened and then passed to a ridge regressor to perform Recursive Feature Elimination (RFE). The reduced set of features obtained from RFE are used to characterize the respective node in the graph. Unlike the feature vectors, the edges are defined using non-imaging phenotypic measures such as age, sex or acquisition site of the fMRI scans (denoted by $M_h$). The function $\gamma$ determines the existence of an edge based on equal phenotypic data. It is defined differently depending on the type of phenotypic measure integrated in the graph. For categorical information such as subject's sex, $\gamma$ is defined as the Kronecker delta function $\delta$, meaning that the edge weight between subjects is increased if e.g. they have the same sex. Constructing edge weights from quantitative measures (e.g. subject's age) is slightly less straightforward. In such cases, $\gamma$ is defined as a unit-step function with respect

---

1. Code available at https://github.com/RichardOlij/Fusing-ss-GCN-for-Autism-Classification

to a threshold. Further details can be found in Section 2.2.2 of Parisot et al. (2018). Using this information about each subject, Parisot et al. (2018) created an adjacency matrix $\mathbf{W}$ of the graph as

$$W_{ij} = Sim(S_i, S_j) \sum_{h=1}^{H} \gamma(M_h(i), M_h(j)). \tag{1}$$

Here, $Sim(S_i, S_j)$ denotes a measure of similarity between the $i^{\text{th}}$ and $j^{\text{th}}$ subject's feature vectors, thus strengthening the links between similar nodes of the graph and weakening the less similar ones. This results in a sparse adjacency matrix.

The primary disadvantage of using non-imaging data such as site information in creating adjacency matrix is the lack of flexibility when scaling to larger datasets, especially if a site has very few subjects or if a new site is added to the database. Moreover, the adjacency matrix defined by Eq. 1 compromises the effectiveness of the GCN architecture. The major advantage of using a GCN is its capability to combine information from two different channels defined on its nodes and edges, respectively. However, due to use of fMRI based similarity measure $Sim(S_i, S_j)$ for defining the edges, there is a significant overlap of resting state fMRI information in $p$-GCN, which eventually limits its discriminative power. To circumvent these issues, the yet unused structural information of the brain can be used to determine the similarities of the brains' structure between subjects. The motivation and advantages of using sMRI information to define connections between subjects is discussed in the subsequent section.

## 3. Proposed Approach

This section provides an overview of our approach. We propose (a) to use the yet unutilized sMRI information of each subject to construct edges between the nodes, (b) define a set of 3D models of the brain using f-MRI information termed as *brain summaries* to extract feature vectors, and (c) fuse these features with the relational information from sMRIs. Details related to these aspects follow below.

### 3.1. Subject similarity using structural MRI

Often the dimensionality of structural MRI data is too high to be directly used for calculating the similarity scores. Thus, a highly compressed version is desired that can still contain sufficient information to derive the extent of similarities between different brain data. To achieve this, we use a pretrained VAE[2] to encode the structural image on to a latent space of significantly lower dimensions (a vector of 200 units). Further, cosine similarity is used to determine the adjacency matrix for all the subjects. Note that unlike the adjacency matrix mentioned in Section 2, the matrix here will not be sparse. Therefore, an automatically determined threshold is applied which ensures that the sparsity is maximized under the constraint that every node on the graph is connected to at least one other node. This has been achieved by setting the threshold to the minimum of all the maximum values of each row (or column). Finally, we create a population graph using feature vectors as in Parisot et al. (2018), but using sMRI information to characterize its edges.

---

2. VAE was trained on the UK Biobank. Information at https://imaging.ukbiobank.ac.uk/.

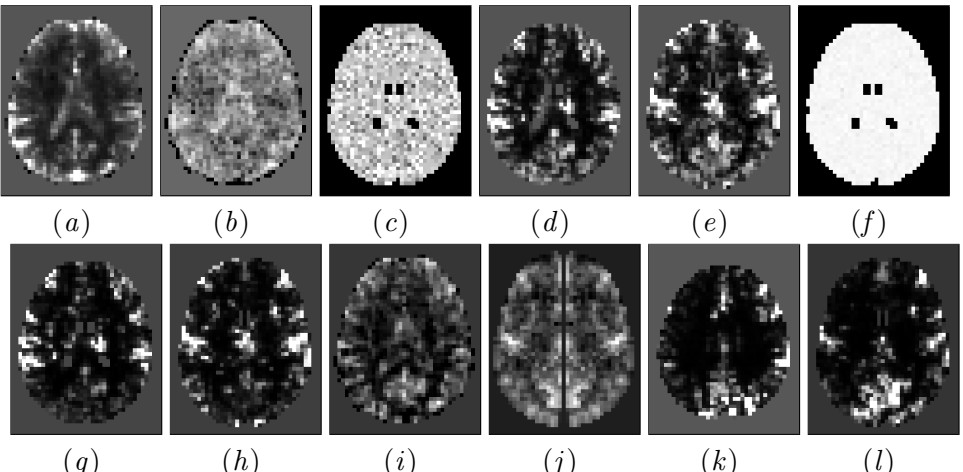

Figure 2: Example slices of 12 brain summaries for a subject from the ABIDE dataset. The summaries are: (a)Amplitude of Low Frequency Fluctuations(ALFF), (b)fractional Amplitude of Low Frequency Fluctuations(fALFF), (c)Autocorrelation, (d)Degree centrality weighted(Dcw), (e)Degree centrality binarize(Dcb), (f)Entropy, (g)Eigenvector centrality binarize(Ecb), (h)Eigenvector centrality weighted(Ecw), (i)Regional Homogeneity(ReHo), (j)Voxel-Mirrored Homotopic Connectivity(VMHC), (k)Local Functional Connectivity Density(LFCD) binarize and (l)Local Functional Connectivity Density(LFCD) weighted

### 3.2. Developing brain summaries

Brain summary refers to a 3D model of the brain obtained by summarizing the information for every voxel along the temporal dimension. To reduce the input dimension, we employ a novel approach of calculating a 'summary' in time for each voxel of the original data, resulting in a 3D volume. Unlike the traditional brain atlases that divide the entire brain into a smaller set of regions, the spatial resolution of the brain image is fully preserved in brain summaries. These measures are informed by the neuroimaging literature, and have been used in the past to study characteristics of the brain image (Zuo et al., 2011; Tomasi and Volkow, 2010; Wang et al., 2014).

In this paper, we employ 12 such summary measures, and investigate their usefulness in the context of autism classification. Example images of these summaries extracted from the ABIDE dataset are shown in Fig. 2. Each summary itself can be interpreted as a biomarker for the rs-fMRI image that provides certain information about a disease state or response to treatment (Davis et al., 2008). In the past, these have been used to reduce the size of time data needed to study a subject for a certain disease and define suitable countermeasures. We employ these summaries in a DL framework and study the influence of each of these on the performance of the model. Our goal is to identify the summaries best suited for GCN models for autism classification. An overview of the summaries is provided in Appendix A.

### 3.3. Fusing sMRI and fMRI data using GCN

The final goal of our work is to combine the relational information from sMRI with the developed brain summaries. A schematic representation of this approach is shown in Figure 1,

and is referred as *ss*-GCN. Since the used brain summaries exist as 3D volumes, these need to be translated to a lower dimensional feature representation so as to be used at the nodes of the graph model. This is done by using a 3D CNN architecture (Ji et al., 2012). The CNN is originally trained on a classification problem of CON or ASD using brain summaries from only the training data. After the model has been trained, the flattened out feature maps from its last layer are treated as feature vectors for the respective brain summaries. The adjacency matrix obtained using sMRI and the feature vectors from brain summaries using fMRI are then fed into the *ss*-GCN model. Additional details related to implementation are described in Appendix D.

## 4. Experiments

We present a series of experiments to demonstrate the effectiveness of *s*-GCN and *ss*-GCN approaches. For this, we use the full dataset from Autism Brain Imaging Data Exchange (ABIDE I and ABIDE II). The ABIDE dataset features 2100+ structural and functional MRI scans of ASD and TD/CON participants from over 30 different acquisition centers. First, we build the baseline model using the approach of Parisot et al. (2018) i.e. *p*-GCN. Further, we present two experiments involving the use of *s*MRI data and brain summaries. Details related to the experimental setups, results and insights follow below.

### 4.1. Comparison of *p*-GCN and *s*-GCN

To demonstrate the advantages of using brain structure over the previously used phenotypic data, we compare the *p*-GCN and *s*-GCN approaches. The motivation behind this experiment is to investigate whether the results obtained with s-GCN, our intermediate model, are comparable to that of the baseline p-GCN. For a fair comparison, we use the same GCN setup as used in Parisot et al. (2018) and perform 10-fold cross validation on the full ABIDE dataset, using their publicly released code[3]. We run the experiments on 9 different atlases namely `AAL`, `cc_200`, `craddock_200`, `HO_cort_maxprob_thr25-2mm`, `JAMA_IC7`, `JAMA_IC19`, `JAMA_IC52`, `schaefer_100` and `schaefer_400`. For all atlases, the same preprocessing procedure and specifications are used. Note that the post-processed atlases used in this paper differ from the atlases in Parisot et al. (2018). However, our experimental set-up for comparison still holds valid since we perform the same pre-processing for both the cases.

    Figure 3 shows the classification performance across 10 folds for *p*-GCN and *s*-GCN obtained on 9 atlases from the full ABIDE database. Note that the performance of s-GCN is lower mostly in the atlases `JAMA_IC7`, `JAMA_IC19` and `JAMA_IC52` which are considered not to be a "true" atlas. We would like to highlight that unlike the rest of the atlas, these atlases are composed of ICA components derived from a previous study on Autism (Cerliani et al., 2015). These are spatially distributed regions and the number of components or regions are rather low [7-52]. It is becoming prudent to give lesser importance to these atlases. Hence, it can be observed that for 5 out of 6 atlases (excluding `JAMA_IC7`, `JAMA_IC19` and `JAMA_IC52`), the mean accuracy values obtained by *s*-GCN are superior to that of *p*-GCN. In general, the average variance across all atlases is lower for *s*-GCN, and we observe that the number of

---

3. Code available at https://github.com/parisots/population-gcn

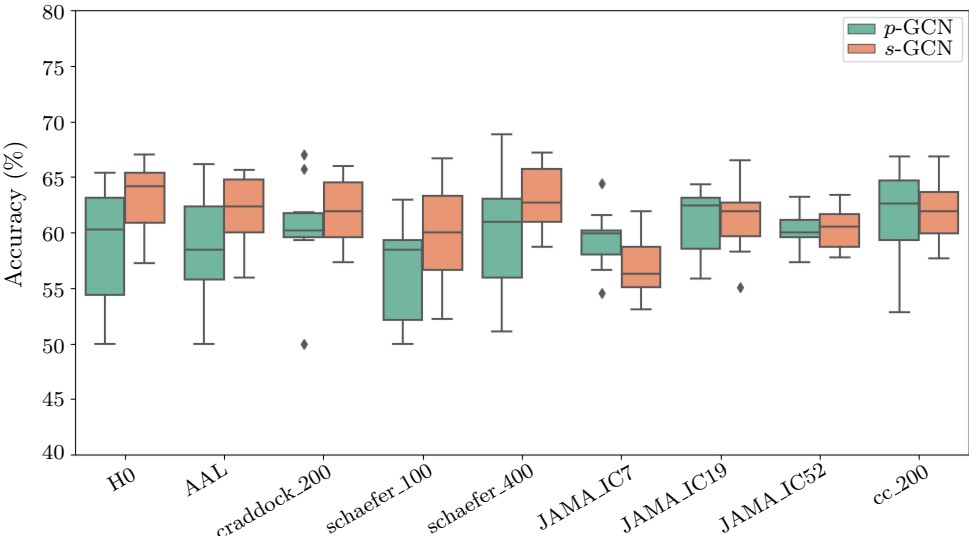

Figure 3: Boxplots denoting performance statistics obtained using $p$-GCN and $s$-GCN on 9 different atlases of full ABIDE dataset.

outliers are significantly reduced. While we see improvements on several sites using $s$-GCN, statistical tests showed that these improvements are close to really significant. Nevertheless, it is assured that $s$-GCN performs at par with $p$-GCN with a smaller standard deviation within the accuracy scores across folds. This implies that replacing the phenotypic data with structural MRI improves the prediction capability of the model as well as its stability.

## 4.2. Using brain summaries in $s$-GCN

To evaluate the use of proposed brain summaries, we replace the features associated with the nodes of the graph in $s$-GCN with the encodings extracted from 3D CNN, as described earlier in Section 3.2. Figure 1 shows the layout of $ss$-GCN; it fuses the fMRI information from brain summaries with the relational information obtained from sMRIs of subjects. We use the same 10-fold cross validation scheme and report the performance of top 5 brain summaries in Table 1. We observe that the performance of top 5 summaries differs by only small margins, with `ReHo`, `fALFF` and `ALFF` performing among the best, followed by `VMHC` and `Dcb`. The reason for `ReHo` performing the best could be attributed to the fact that it is a more stable voxel-based measure and is relatively more sensitive compared to the other measures (Chen et al., 2017). Also, as expected, `fALFF` performs slightly better than `ALFF` due to its reduced sensitivity to physiological noise. Results of all the 12 summaries are reported in Appendix E.

## 4.3. Across sites cross validation

The ABIDE dataset is aggregated from multiple sites using different scanner types and acquisition protocols such as scanning time and repetition time (TR). Hence, the dataset contains sensitive variations that compromise the consistency between sites. To reduce the

| Brain Summary | Accuracy (in %) | AUC |
|---|---|---|
| ReHo | 62.6±2.7 | 0.68±0.02 |
| fALFF | 61.8±2.5 | 0.65±0.03 |
| ALFF | 61.2±3.8 | 0.66±0.04 |
| VMHC | 60.9±2.9 | 0.63±0.04 |
| Dcb | 60.9±2.2 | 0.64±0.02 |

Table 1: Mean accuracy and AUC with standard deviation for top 5 performing brain summaries obtained using ss-GCN

| Acquisition Site | p-GCN | s-GCN | ss-GCN |
|---|---|---|---|
| ABIDEII-GU_1 | 59.1±4.4 | 59.7±3.0 | **68.0**±2.3 |
| ABIDEI-USM | 60.9±4.5 | 60.9±2.9 | **61.1**±2.3 |
| ABIDEI-UM_1 | 59.4±5.6 | **62.3**±3.4 | 61.5±2.7 |
| ABIDEII-KKI_1 | 50.1±4.1 | 50.8±3.2 | **68.9**±2.1 |
| ABIDEI-NYU | **65.3**±3.5 | 64.5±3.4 | 63.0±2.6 |

Table 2: Mean test accuracy ± standard deviation over 10 test-train splits of different GCN models for the leave-one-site-out experiment

effect of site-specific sources of variability and assess the robustness of the classification model, leave-one-site-out cross-validation experiments are performed. The left out site for every training process is used as the test set to evaluate the model. The motivation for designing such an experimental setup is to test adaptability of the model to previously unseen sites. Therefore, we perform leave-one-site-out experiment to compare p-GCN with the proposed s-GCN and ss-GCN approaches. For an equitable comparison between the models, we choose the best performing atlases for p-GCN and s-GCN as shown in Figure 3 (cc_200 and H0 respectively), and the best summary for ss-GCN as in Table 1 (ReHo).

We report the accuracy scores on 5 sites that contribute most to the number of subjects in ABIDE. Details related to the subject composition from various sites can be found in Appendix B. It should be noted that data from the same acquisition center but different ABIDE collection (I or II) are treated as being from different sites. This is because data in ABIDE-II from the same center can have different scanning protocols, repetition time (TR) and so on. Table 2 gives the accuracy scores (with the standard deviation) of the leave-one-site-out experiment on these 5 sites. For 4 out of 5 sites, our proposed approaches outperform the baseline p-GCN method. In particular, we observe that for the sites ABIDEII-KKI_1 and ABIDEII-GU_1, our ss-GCN approach provides remarkable improvements over the p-GCN method of 18.8% and 8.9%, respectively. The large variations in the performances across sites is a result of vast heterogeneity in datasets between sites. These occur either due to many reasons such as different SNR per site (as can be referred from Appendix C) and different MRI image acquisition parameters at every site. Now, s-GCN has elements of the cross-correlation matrix as input compared to ss-GCN that have the brain summaries, thus the effect of these heterogeneities could be dramatically different. Moreover, it can be seen that the variance in results of ss-GCN and s-GCN is much lower than p-GCN. This shows the robustness and generalizability of our proposed model in classifying Autistic (ASD) and healthy controls (CON) across multiple sites.

## 5. Conclusions

In this paper, we utilize relational information from sMRI data as compared to phenotypic data together with fMRI data for autism classification using GCNs. Our results on full ABIDE dataset demonstrate that the proposed approach performs better than that using phenotypic data. Further, we show that replacing the atlases with brain summaries makes the model more robust for new sites with the best case improvement exceeding 18%. Unlike

the previous works, we show that the model can perform well even without subjectively picking samples from the full dataset. This implies that our model generalizes well under scenarios of higher noise levels. To conclude, our *ss*-GCN model, operating on structural and functional MRI data and using brain summaries, performs at par or above the conventional approach for autism classification. In another preliminary experiment, we explicitly combined the classifiers from all brain summaries by making a voting classifier per fold. The result did seem to improve with a mean accuracy of 64.23% and mean AUC score of 68.31%. It would be of interest for the neuroimaging community to consider it as an alternate direction of research for autism classification using GCNs.

## Acknowledgments

GW received funding from Philips Research for another research project.
RMT was supported by the Netherlands Organization for Scientific research (NWO/ZonMw Vidi 016.156.318).

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

# Appendix A. Brain Summaries

Table 3: Description of the 12 brain summaries.

| Summary | Explanation |
|---|---|
| ALFF | Amplitude of Low Frequency Fluctuations (ALFF) measures spontaneous fluctuations in the Blood Oxygenation Level Dependent (BOLD) fMRI signal of a certain region in the resting brain (Zou et al., 2008). BOLD stands for the ratio between the oxygen-rich and poor haemoglobin. |
| Autocorr | Autocorrelation (autocorr) is the correlation of a signal with a delayed copy of itself, used to find repeating patterns and finding the dominant frequency of a signal (Olszowy et al., 2019). fMRI data is positively autocorrelated in time, which is a result of neural sources, scanner induced low-frequency drifts, respiration and cardiac pulsation. This results in noise what can result in false positives during classification tasks, therefore there are packages, like AFNI (Cox, 1996), for fMRI research that reduce this autocorrelation noise. |
| Degree centrality binarize (Dcb) | Degree centrality is a measure of local network connectivity and identifies the most connected nodes by counting the number of direct connections to all other nodes (Zuo et al., 2011). Degree centrality analysis emphasizes higher order cortical association areas while showing reduced sensitivity for paralimbic and subcortical regions. Binarizedholds that the connection strength is either 0 or 1.**** |
| Degree centrality weighted (Dcw) | Same as above the only difference is that for weighted the connections strength is a correlation value.**** |
| Eigenvector centrality binarize (Ecb) | Eigenvector centrality is a measure of global network connectivity it holds that the higher the eigenvector centrality for a node, the more connections the node has with other nodes that have high centrality (Zuo et al., 2011). Eigenvector centrality is more sensitive to paralimbic and subcortical regions, which is in contrast to degree centrality . Binarized means that the connection strength is either 0 or 1.**** |
| Eigenvector centrality weighted (Ecw) | Same as above only difference is that weighted stands for that the connection strength is a correlation value.**** |
| Entropy | Entropy indicates the irregularity within a system, which remains relativly low in living systems but increases over time in any closed system, such as our universe (Sandler, 2017). The human brain is the most complex living organism known to men, therefore, it has a prominent need to sustainentropy to function properly (Singer, 2009). Since fMRI measures regional changes in brain blood flow and metabolism it is a good measure for entropy (Wang et al., 2014). |
| fALFF | Fractional alff (fALFF) reduces the sensitivity of ALFF to physiological noise by taking the ratio of each frequency (0.01-0.08 Hz) with relation to the total frequency range (0-0.25 Hz) (Zou et al., 2008) |
| LFCD binarize | Local Functional Connectivity Density (LFCD) is also a measure of local network connectivity. Unlike degree centrality and eigenvector centrality which can be calculated for Regions of Intrest (ROIs)*, LFCD needs a voxel-based** mask (Tomasi and Volkow, 2010). It then finds a mapping from the given mask to its neighbours and so on until the connections become weaker than a specified threshold. Binarized means that the connection strength is either 0 or 1.**** |
| LFCD weighted | Same as above only weighted means that the connection strength is a correlation value.[4] |
| ReHo | Regional Homogeneity (ReHo) is a voxel-based measure that evaluates the similarity between the time series of a given voxel and its nearest neighbours (Zang et al., 2004). The measure is based on the hypothesis that intrinsic brain activity is embodied by clusters of voxels rather than single voxels |
| VMHC | Voxel-Mirrored Homotopic Connectivity (VMHC) quantifies functional homotopy by providing a voxel-wise measure of connectivity between the hemispheres*** (Zuo et al., 2010). This is done by computing the connectivity between each voxel in one hemisphere and it's mirrored counterpart in the other |

\* A Region of Interest (ROI) is a subset of the fMRI image identified to be of any particular purpose (Poldrack, 2007)

\** A voxel is a pixel representing a value in the three-dimensional space, that corresponds to a pixel for a given slice thickness
https://blogs.scientificamerican.com/observations/whats-a-voxel-and-what-can-it-tell-us-a-primer-on-fmri

\*** Hemispheres are the two halves of the that together form the brain and are separated by a deep groove

\**** Network centrality - https://fcp-indi.github.io/docs/user/centrality.html

## Appendix B. ABIDE I-II Dataset

Table 4: Distributions of Autistic (ASD) and healthy control (CON) subjects per site and dataset. The entries in bold refers to the top-5 sites with respect to total number of subjects.

| Acquisition Center | Site | Autistic | Control | Total |
|---|---|---|---|---|
| California Institute of Technology | ABIDE-I_CALTECH | 19 | 19 | 38 |
| Carnegie Mellon University | ABIDE-I_CMU | 13 | 14 | 27 |
| Kennedy Krieger Institute | ABIDE-I_KKI | 33 | 22 | 55 |
| | **ABIDE-II_KKI1** | **155** | **56** | **211** |
| Katholieke Universiteit Leuven | ABIDE-I_LEUVEN_1 | 15 | 14 | 29 |
| | ABIDE-I_LEUVEN_2 | 20 | 15 | 35 |
| | ABIDE-II_KUL_3 | 0 | 28 | 28 |
| Ludwig Maximilians University Munich | ABIDE-I_MAX_MUN | 33 | 24 | 59 |
| | **ABIDE-I_NYU** | **105** | **79** | **184** |
| New York University | ABIDE-II_NYU_1 | 30 | 48 | 78 |
| | ABIDE-II_NYU_2 | 0 | 27 | 27 |
| Oregon Health and Science University | ABIDE-I_OHSU | 15 | 13 | 28 |
| | ABIDE-II_OHSU_1 | 54 | 37 | 91 |
| Olin, Institute of Living at Hartford Hospital | ABIDE-I_OLIN | 16 | 20 | 36 |
| | ABIDE-II_OILH_2 | 35 | 24 | 59 |
| University of Pittsburgh | ABIDE-I_PITT | 27 | 29 | 56 |
| Social Brain Lab | ABIDE-I_SBL | 15 | 15 | 30 |
| San Diego State University | ABIDE-I_SDSU | 22 | 14 | 36 |
| | ABIDE-I_SDSU_1 | 25 | 33 | 58 |
| Stanford University | ABIDE-I_STANFORD | 20 | 20 | 40 |
| Trinity Centre for Health Sciences | ABIDE-I_TRINITY | 25 | 24 | 49 |
| | ABIDE-II_TCD1 | 21 | 21 | 42 |
| University of California Los Angeles | ABIDE-I_UCLA_1 | 32 | 41 | 73 |
| | ABIDE-I_UCLA_2 | 13 | 13 | 26 |
| | ABIDE-I_UCLA_1 | 16 | 16 | 32 |
| University of Michigan | **ABIDE-I_UM_1** | **55** | **55** | **110** |
| | ABIDE-I_UM_2 | 22 | 13 | 35 |
| University of Utah School of Medicine | **ABIDE-I_USM** | **43** | **58** | **101** |
| | ABIDE-II_USM_1 | 26 | 17 | 43 |
| Barrow Neurological Institute | ABIDE-II_BNI_1 | 29 | 29 | 58 |
| Erasmus University Medical Center | ABIDE-II_BNI_1 | 27 | 25 | 52 |
| ETH Zürich | ABIDE-II_ETH_1 | 24 | 13 | 37 |
| Georgetown University | **ABIDE-II_GU_1** | **55** | **51** | **103** |
| Institut Pasteur and Robert Debré Hospital | ABIDE-II_IP_1 | 34 | 22 | 56 |
| Indiana University | ABIDE-II_IU_1 | 20 | 20 | 40 |
| San Diego State University | ABIDE-II_SDSU_1 | 25 | 33 | 58 |
| University of California Davis | ABIDE-II_UCD_1 | 14 | 18 | 32 |
| University of Miami | ABIDE-II_U_MIA_1 | 15 | 13 | 28 |

## Appendix C. Temporal Signal to Noise Ratio across sites

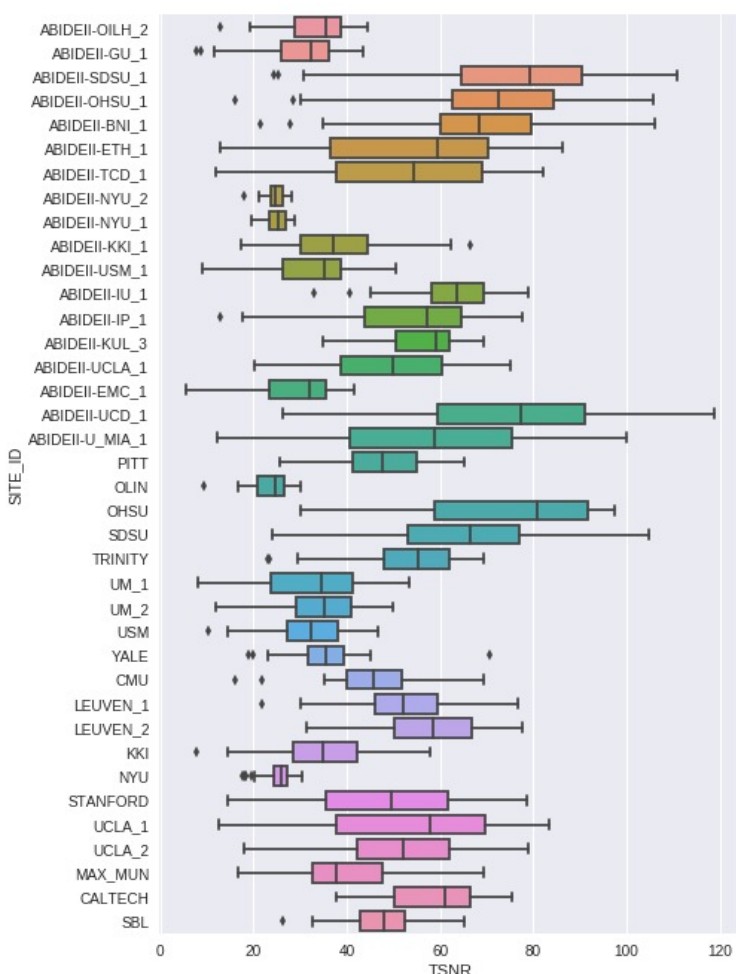

Figure 4: Temporal Signal to Noise Ratio (TSNR) per acquisition site

## Appendix D. Implementation Details

This section provides brief details related to the implementation of the GCN and 3D CNN. These follow in the subsections below.

### D.1. Graph Convolutional Network

The network uses two hidden layers with 16 units in each layer. The learning rate and dropout values of 0.0005 and 0.3 are used, respectively, and the network is optimized for 500 epochs. For approximating the convolutions, several different values of Chebyshev polynomial order $K$ were tested, and we found $K = 3$ to be best suited.

### D.2. Feature Extraction from brain summaries using 3DCNN

Figure 5 shows a schematic representation of the model architecture, comprising dimensions of $45 \times 54 \times 45$. In total, two convolutional layers followed by max poolings are used. The sets of filters used for every convolutional layer are described in Figure 5. The output from the second convolutional layer passes through two fully connected layers. The entire architecture is trained for classification of subject's brains. Additional hyperparameters such as learning rate and momentum are set to 0.001 and 0.9, respectively. The network is trained for 300 epochs with batch size of 32.

After the network is trained, the intermediate output of 3000 dimensional size is then used as one of the inputs to the GCN.

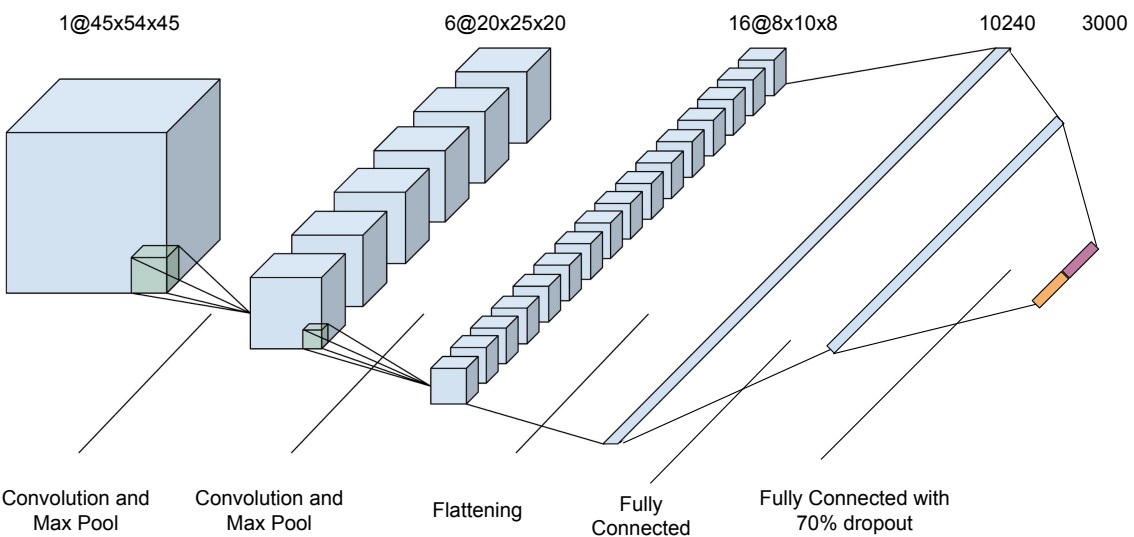

Figure 5: Illustration of 3DCNN architecture.

## Appendix E. Additional Results

| Brain Summaries | Acc.(%) | AUC |
|---|---|---|
| ReHo | 62.6 | 0.72 |
| fALFF | 61.8 | 0.71 |
| ALFF | 61.2 | 0.71 |
| VMHC | 60.9 | 0.69 |
| Dcb | 60.9 | 0.70 |
| Dcw | 60.1 | 0.68 |
| Ecb | 59.3 | 0.67 |
| Ecw | 59.2 | 0.65 |
| LFCD binarize | 58.1 | 0.61 |
| LFCD weighted | 57.6 | 0.61 |
| Entropy | 54.3 | 0.53 |
| Autocorr | 51.2 | 0.51 |

Table 5: Mean classification accuracy and average Area Under the Curve (AUC) scores across 10-folds for all the 12 brain summaries obtained using *ss*-GCN.

