# OpenReview forum: "Fusing Structural and Functional MRIs using Graph Convolutional Networks for Autism Classification"
_MIDL.io/2020/Conference — MIDL 2020_

### Official Review · AnonReviewer4 · 2020-03-11
**Strong paper, sound results**

**Rating:** 4
**Confidence:** 5
**Recommendation:** Oral, Poster

**Summary:**

This is a strong paper which extends recent work on using graph convolutional networks to model phenotypic variation from fMRI. The paper builds from Parisot et al 2017 to build a population graph which uses similarity of latent encodings of structural MRI to create the graph structure. Nodes in the graph are represented from summaries of fMRI data compressed using the 3D CNN. The paper shows variable improvement over Parisot when just the graph structure is modified with sMRI but in some cases the combined and use of sMRI and fMRI encodings are shown to give good improvement for Autism prediction from data collected from different sites.


**Strengths:**

The paper presents a new method for graph convolutional learning from brain MRI, which fully leverages the advantages of graph CNNs to combine sMRI and fMRI. This is important as behavioural and cognitive phenotypes are likely to originate from morphological and dynamic functional sources.

It presents a method which is independent from the need to define ROIs from an atlas. This is shown in Figure 3 to lead to very variable performance for the functional connectivity based approach (Parisot 2017)

The approach is well motivated and evaluated. Methods are generally clear.


**Weaknesses:**

Figure 3 isn't wholly convincing as you would imagine that users would always want to select the atlas which corresponds to the best performing model.

The two step validation of the changes relative to Parisot et al are not immediately clear. It should be made more obvious that 4.1 refers just the changing the way in which the graph edge structure is learnt and thus both networks are using functional connectivity for nodes at that point.



**Detailed Comments:**

This is a strong paper which extends recent work on using graph convolutional networks to model phenotypic variation from fMRI.

Several works have indicated that variation in functional connectivity is likely driven by morphological variation so in that sense it makes sense to go to the source and use sMRI to model the edge structure of the graph. Then model fMRI temporal variance in the nodes.

Figure 3 isn't wholly convincing as you would imagine that users would always want to select the atlas which corresponds to the best performing model.  Nevertheless figure 4 shows very promising results.  It would be interesting to dig into the reasons for the significant improvement for sites ABIDEII-KKI 1 and ABIDEII-GU 1 to know what is driving this

It would be interesting to know why the authors didn't  use a combination of scalar measures od function.  Was it a computational bottleneck or tried and shown not to work?

Minor points

There are no References  of links to the atlases.

The structure of the paper could be better. I'm not sure it should be necessary to go to the appendix to find which type of  graph CNN was used. I'm not sure the section headings (3.1-3) summarise well the key components of the network,


**Justification Of Rating:**

The methods in this paper are novel and well motivated. It's thoroughly validated against a competing method and is shown to offer improvement. The paper shows highly novel combination of sMRI and fMRI for phenotype prediction.

**Paper Type:**

both

**Special Issue:**

no

---

> ### Author Response · Authors · 2020-03-28
> **Response to Reviewer 4**
>
> We thank you for the comments.  These comments were helpful in improving the final draft, and below we provide our response.
>
> Further inspection of results:
> We will modify the statement to reflect the observation of the reviewer. The results from p- and s-GCN are comparable with s-GCN showing a marginal improvement at best. In the updated manuscript we will provide the results of a statistical test to quantify these observations. Also, we would like to iterate that s-GCN was a stepping stone towards our final model - the ss-GCN - which according to Table-2 has a much better performance. We will add a discussion related to the analysis of edges for the population graphs of p-GCN and s-GCN based on  (i) Overall degree distribution, (ii) Degree distribution split according to class.  For a better measure of classification performance, we now also report balance accuracy score and area under the ROC curve (AUC). These details are added in Section 4 of the updated draft.
> Note: The performance of s-GCN is lower mostly in the atlases we called JAMA-x in Fig.3. We would like to highlight that this is not a “true” atlas like the rest of them and is composed of ICA components derived from a previous study on Autism [Cerliani et al., JAMA Psychiatry, 2015]. These are spatially distributed regions and the number of components or regions is rather low [7-52]. It is becoming prudent in the community to give lesser importance to these “atlases”. We will add this remark in the updated draft.
>
> Combination of scalar performance measures:
> Based on your suggestion, we first performed an experiment where we explicitly combined the classifiers from all brain summaries by making a voting classifier per fold. The result did seem to improve with a mean accuracy of 64.23% and AUC score of 68.31%. We will add a detailed discussion of this experiment in our updated draft.
> As per as combining them implicitly as input, we used a 3D CNN with brain summaries as channels to access the performance of using all summaries simultaneously. Unfortunately, this experiment did not yield good results compared to many of the individual summaries. Also, performing a full CV using all brain summaries as channels would have been computationally intensive. Thus, we omitted the experiment from the paper. But, due to several requests by the reviewers, we shall include a discussion on this experiment in the discussion section of the paper.
>
> Why some sites are better:
> We provided a new figure in the appendix that shows the SNR of the different sites. ABIDE-II KKI1 and ABIDE-II GU-1 do not seem to have a better SNR than the rest and therefore cannot be the reason for the difference. In order to fully address this question, we will have to perform further analysis and visualization which unfortunately is beyond the scope of our current paper.
>
> References to atlases:
> The references of the atlases will be added and the atlases themselves will be made available in the GitHub repo.
>
> Paper structure:
> We will make a few changes to the structure in the final version of the paper.
>
> References:
> [1] Cerliani, L., Mennes, M., Thomas, R. M., Di Martino, A., Thioux, M., & Keysers, C. (2015). Increased functional connectivity between subcortical and cortical resting-state networks in autism spectrum disorder. JAMA psychiatry, 72(8), 767-777

---

### Official Review · AnonReviewer1 · 2020-03-17
**Interesting multimodal approach to multi-site autism classification with sensitivity analysis**

**Rating:** 3
**Confidence:** 4

**Summary:**

The authors extend previous work by Parisot et al to include structural MRI similarity or various ‘brain summaries’ for functional MRI. Structural MRI similarity is captured by a 200-dimensional latent representation from a pretrained VAE, while brain summaries are represented by a 3000-dimensional feature map from a CNN trained on controls vs ASD classification. Structural similarity from low-dim representations is used to compute edge weights for an adjacency matrix between subjects, where vertices are subjects, while low-dim brain summaries are used as vertex feature vectors. A graph convnet then classifies unlabeled vertices (subjects). Performance is compared using various atlases, brain summaries, and removing specific sites.

**Strengths:**

-	Using open data throughout (ABIDE I/II), in particular multi-site data, is a great strength of the paper, enabling approximate comparison with previously published work.
-	Classification accuracy seems slightly higher than in earlier work (e.g. on ABIDE I with fMRI Nielsen et al. Front. Hum. Neurosci 2013, or with sMRI Haar et al. Cerebral cortex 2014, Sabuncu et al. Neuroinform 2015). Multi-site data is difficult to handle and neuropsychiatric disorders are difficult, so this is a good point.
-	Evaluating performance of functional connectivity-based measures on multiple atlas is very important (Dadi et al. NeuroImage 2019) and is a strong point here.
-	Likewise, comparing performance with leave-one-site out is a great idea (but see below).


**Weaknesses:**

The main weakness is the statistical reporting, in particular in terms of performance metrics and classification confounders such as site, class balance, or sex balance. The focus is almost exclusively on comparison why the baseline from Parisot et al, with little regard to clinical features.

-	The dataset is imbalanced in terms of class. With high enough imbalance this can drive classification purely by virtue of class frequencies. Authors should compute and show diagnosis proportions (in total and at each cross-val fold), and report on the no-information rate each time. Authors should also report at least one metric measuring differential performance between classes, such as F1, Kappa, or both sensitivity and specificity, rather than overall accuracy.
-	In Tables 1 and 2, no estimate of standard deviation across folds is provided. It is hard to judge if the differences are significant or not. Are these figure computed on the re-assembled confusion matrix after the 10 folds, or averaged from estimates in each fold? Comparison with figure 3 makes this confusing.
-	Likewise, the male-to-female ratio for autism is around 3:1 (Loomes et al., JAACP, 2017). Depending on data split this alone can drive classification. The authors should report on male-female ratio in their data. A baseline should also be provided with a basic phenotype-based classifier (e.g. random forest with Sex and Age (possibly site)) as input.
-	For the multi-site study: the distinction between acquisition center and site (appendix B) is not clear and probably muddies the generalization claims. In 4/5 “left-out sites”, the authors in fact use data from the same acquisition center in the training set. Only Georgetown is cleanly split. At present it is hard to conclude as to generalization ability.
-	The motivations and process behind important hyperparameter choices (gamma in equation 1 across all methods, VAE latent space dimension, CNN last layer feature map dimension) are not explained and the tuning process is not explained. How do we know this is not done looking at overall results?
-	It is not clear whether the CNN is retrained at each cross-validation fold from brain summaries


**Detailed Comments:**

- « brain hardware » is awkward phrasing (also implying there is “brain software”, philosophically complicated) and should be replaced by « brain ».

**Justification Of Rating:**

Interesting, but complicated and somewhat over-engineered approach for fusing structural MRI and functional connectivity. Nevertheless sensitivity analysis is nice (with caveats) and some improvement is shown over the baseline method.

**Paper Type:**

both

**Questions To Address In The Rebuttal:**

-	Experiment with cleaner site splits. Rather than treating data from same acquisition center but different collection phase as a different it would be better to completely remove an acquisition center rather than just removing one dataset per site
-	See also above for suggestions


**Special Issue:**

no

---

> ### Author Response · Authors · 2020-03-28
> **Response to Reviewer 1**
>
> We thank you for the comments.  These comments were helpful in improving the final draft, and below we provide our response.
>
> Performance metrics and handling class imbalance:
> We have ensured that our results are not biased due to class imbalance by using “random stratified k-fold splits” which splits the data in such a way that labels are balanced. Stratification is the process of rearranging the data to ensure each fold is a good representative of the whole. In our experimental setup, we use k=10 and report the mean accuracy value. The stratified k-fold function ensures that in every batch the distribution of classes is the same and cross-validation ensures that the model does not overfit. For a better measure of classification performance, we now also report balance accuracy score and area under the ROC curve (AUC). These details are added in Section 4 of the updated draft.
>
> Reporting in Tables 1 and 2:
> We report the mean accuracy results across 10 folds in Table 1 and Table 2. Based on your feedback, we have reported the standard deviation with the accuracy scores in the final draft of the paper.
>
> Male to female ratio and baseline performance:
> The reviewer raises an important point about the skewed male to female ratio in the patient population and thus the possibility of gender driving the classification. To assess this, we actually built a CNN that is trained to classify males vs females in the UK Biobank dataset. We then used that network on ABIDE to classify ASD vs CON, and the results were very poor — indicating that gender features alone in the scan did not drive the classification.
> A detailed comparison of GCN based model with classifiers such as a multi-layer perceptron (MLP), random forest, Ridge classifier, and autoencoder was performed already in Parisot et.al. 2018. They also compared various combinations of non-imaging variables coupled with GCN. Hence, in our work, we took the best performing model by Parisot et.al. 2018 as our baseline approach (p-GCN) and evaluated our models with this baseline model.
>
> Generalization for acquisition center and site:
> In order to perform leave-one-site-out experiments we treat data from ABIDE-I and ABIDE-II  differently even though they might be acquired at the same center. This is because ABIDE-II in the same center can have different scanning protocols, repetition time (TR) and so on. We have provided a new figure now in our paper that shows the SNR of the image acquisitions in all sites. You would see that ABIDE-II has a different distribution compared to ABIDE-I. Due to these differences, it is safe to assume that the dataset in ABIDE-I and -II can be treated differently. We will, however, report results from different acquisition centers as well in our final draft.
>
> Choice of hyperparameters and tuning choice:
> Gamma is not used as a hyperparameter, it is a fixed value for each type of phenotypic data. For defining graphs in p-GCN we used the exact same procedure as used in our baseline approach Parisot et al. (2018), where “Gamma” is defined differently depending on the type of phenotypic measure integrated in the graph. For categorical information such as subject’s sex, they define “Gamma” as the Kronecker delta function δ, meaning that the edge weight between subjects is increased if e.g. they have the same sex. Constructing edge weights from quantitative measures (e.g. subject’s age) is slightly less straightforward. In such cases, they defined “Gamma” as a unit-step function with respect to a threshold. Further details can be found in Section 2.2.2 of Parisot et al. (2018).
> In deep learning, the general workflow involves starting from a “standard” model (for example a VGG or Resnet) architecture. If the results are not at par, the experimenter varies some hyperparameters like the number of FC layers and the number of neurons in those layers. Consider this akin to a random grid search. However debatable, in the ML community, this is often overlooked when reporting the performance because given the number of parameters (>10M) the performing any sort of grid search would be prohibitive. Added to that typically, we do not adjust the architecture on the entire dataset but on a subset and then we freeze the architecture for the rest of the experiments.
>
> Training of CNN:
> No, 3DCNN is not retrained for each cross-validation. The 3DCNN is however trained separately for each brain summary and the final output is saved as feature vectors for each of these summaries.
>
> We will take your comments on the use of the word "hardware" with brain and further in future will be careful in not using such words in conjecture with brain.
>
> References:
> [1] Sarah Parisot, Sofia Ira Ktena, Enzo Ferrante, Matthew Lee, Ricardo Guerrero, Ben Glocker, and Daniel Rueckert. Disease prediction using graph convolutional networks: Application to autism spectrum disorder and alzheimer’s disease. Medical image analysis, 2018

---

### Official Review · AnonReviewer2 · 2020-03-19
**Fusing Structural and Functional MRIs using Graph Convolutional Networks for Autism Classification**

**Rating:** 3
**Confidence:** 5
**Recommendation:** Poster

**Summary:**

This paper presents a graph neural network-based method for autism classification using structural and functional fMRI. Specifically,  each subject is defined as a node of the graph,  the sMRI's feature similarity is used for building node connection and the brain summarizes are used to build the node feature. The proposed method is tested on the ABIDE dataset.

**Strengths:**

-The paper is overall clearly written and easy to follow.
-The idea of using sMRI feature similarity between subjects to build the edge of the population graph is novel in neuroimaging analysis.
-The results are tested on different hyper-parameter choices.

**Weaknesses:**

- The motivation for using sMRI similarity. Do autism subjects have similar sMRI? Does the scanner affect the measure, as different scanners were used in different sites? It would be good to see the supporting references or statistic analysis and investigating the difference /relationship of the edges between the graphs in p-GCN and the graphs in s-GCN.
- Compared with p-GCN, the proposed s-GCN seems not significantly improve classification accuracy.
- Not obvious to validate the statement 'In general, the average variance across all atlases is lower for s-GCN' from Fig 3.






**Justification Of Rating:**

The paper is built on existing work (Parisot et al. 2018). The extension seems novel. However, the correctness and motivation of using sMRI to construct edges need further justification.  The accuracy is not exciting compared to the other existing works.

**Paper Type:**

methodological development

**Questions To Address In The Rebuttal:**

see weakness

+ Will using the concatenation of all brain summaries as node feature will improve the final results?

**Special Issue:**

no

---

> ### Author Response · Authors · 2020-03-28
> **Response to Reviewer 2**
>
> We thank you for the comments.  These comments were helpful in improving the final draft, and below we provide our response.
>
> Motivation for using sMRI:
> Previous studies ( Su et al., 2012, Tyan et al., 2017 and Littmann et al., 2006) have shown associations between sMRI and age/sex/site. These studies also indicate, for example, that “brain-age” need not always coincide with the age reported. In order to avoid such uncertainties in establishing the edges on a graph, we resorted to the use of sMRI images directly —  giving us one variable to establish the relationship.
>
> Effect of scanners/sites:
> Every site has its own MRI-scanner and this could be from different manufacturers. Also, each site has its own acquisition parameters; resolution, the direction of acquisition, and so on. All these differences are imprinted in the sMRI images. Therefore, finding a relationship (edge probability) between two subjects is more semantically meaningful in the space of images as opposed to in a vector of mixed continuous and discrete parameters.
> Analysis of the edges:
> We will add a discussion related to the analysis of edges for the population graphs of p-GCN and s-GCN based on  (i) Overall degree distribution, (ii) Degree distribution split according to class.
>
> Improvement with s-GCN:
> We will modify the statement to reflect the observation of the reviewer. The results from p- and s-GCN are comparable with s-GCN showing a marginal improvement at best. In the updated manuscript we will provide the results of a statistical test to quantify these observations. Also, we would like to iterate that s-GCN was a stepping stone towards our final model - the ss-GCN - which according to Table-2 has a much better performance. We will add a discussion related to the analysis of edges for the population graphs of p-GCN and s-GCN based on  (i) Overall degree distribution, (ii) Degree distribution split according to class.  For a better measure of classification performance, we now also report balance accuracy score and area under the ROC curve (AUC). These details are added in Section 4 of the updated draft.
> Further, the focus of our work was to eliminate the need of using phenotypic information because:
> (a) using non-imaging data such as site information in creating an adjacency matrix reduces the transferability of the model to access new sites, as we would not know how to “connect” these patients to the graph.
> (b)  Anonymizing structural scans are easier and most protective of privacy than using phenotypic data.
>
> Note: The performance of s-GCN is lower mostly in the atlases we called JAMA-x in Fig.3. We would like to highlight that this is not a “true” atlas like the rest of them and is composed of ICA components derived from a previous study on Autism [Cerliani et al., JAMA Psychiatry, 2015]. These are spatially distributed regions and the number of components or regions is rather low [7-52]. It is becoming prudent in the community to give lesser importance to these “atlases”. We will add this remark in the updated draft.
>
> Combining all brain summaries:
> Based on your suggestion, we first performed an experiment where we explicitly combined the classifiers from all brain summaries by making a voting classifier per fold. The result did seem to improve with a mean accuracy of 64.23% and AUC score of 68.31%. We will add a detailed discussion of this experiment in our updated draft.
> As per as combining them implicitly as input, we used a 3D CNN with brain summaries as channels to access the performance of using all summaries simultaneously. Unfortunately, this experiment did not yield good results compared to many of the individual summaries. Also, performing a full cross validation using all brain summaries as channels would have been computationally intensive. Thus, we omitted the experiment from the scope of this paper.
>
> References:
> [1] Longfei Su, Lubin Wang, and Dewen Hu. Predicting the age of healthy adults from structural mri by sparse
>  representation. In International Conference on Intelligent Science and Intelligent Data Engineering, pages 271–279. Springer, 2012.
> [2] Yeu-Sheng Tyan, Jan-Ray Liao, Chao-Yu Shen, Yu-Chieh Lin, and Jun-Cheng Weng. Gender differences in the structural connectome of the teenage brain revealed by generalized q-sampling mri. NeuroImage: Clinical, 15:376–382, 2017.
> [3] Arne Littmann, Jens Guehring, Christian Buechel, and Hans-Siegfried Stiehl. Acquisitionrelated morphological variability in structural mri. Academic radiology, 13(9):1055–1061, 2006.
> [4] Cerliani, L., Mennes, M., Thomas, R. M., Di Martino, A., Thioux, M., & Keysers, C. (2015). Increased functional connectivity between subcortical and cortical resting-state networks in autism spectrum disorder. JAMA psychiatry, 72(8), 767-777

---

### Official Review · AnonReviewer5 · 2020-03-20
**Review for "Fusing Structural and Functional MRIs using Graph Convolutional Networks for Autism Classification "**

**Rating:** 3
**Confidence:** 4

**Summary:**

In this work, the authors have proposed a GCN framework for autism detection. The novetly in the work involves
a) using the structural MR features from a VAE to determine the edge weights.
b) Considering the effect of various methods for temporal brain summaries from FMRI to be used in node features


**Strengths:**

I believe that the contributions are novel. The idea of exploring explicit structural information as edge features, instead of high-level proxies, is interesting. The results, especially when considering brain summaries are good.
(However, please see weaknesses section, calling for more discussion on this)

**Weaknesses:**

Based on some of the results, the effectiveness of the primary contribution seems (of adding structural information) to be mixed, and some parts of the paper need to be improved. Please see my comments below:

a) In the comparison of p-GCN and s-GCN, in Fig. 3 and Table 2, there are small differences between most cases of p-GCN and s-GCN, and in some cases p-GCN ourperforms s-GCN. Thus, only adding structural information does not yield consistent results. Only when the brain summary is also included as a part of ss-GCN, one can notice a good improvement (in some cases). Thus, I am curious how the performance would be if brain summaries are used in conjunction with p-GCN.

b) Also, the improvement with ss-GCN in Table 2, is very different across cases. Please discuss such a large variation in the results with ss-GCN.

c) The statement on Page 3, "Hence, as opposed to defining relations ... expected to have lower variance", is not clearly elaborated. More specifically, what do the authors imply that the strcutural representations have lower variance. I am assuming that it is the within-class variance that they mean. However, it is not clear. Please elaborate, and justify.

d) The use of gamma in equation (1) seems like a function, rather than a weight (as mentioned in the paragraph below the equation), as M_h(i) and M_h(j) seem to be arguments of the functions. Please clarify / correct this.

e) In section 3.1, the authors mention that they use a pre-trained VAE. Please specify what data is used for pre-training, and why is relevant for this application.

f) Perhaps, the authors can also compare with a couple of other contemporary methods which also obtained similar results.
(e.g. Distance Metric Learning using Graph Convolutional Networks: Application to Functional Brain Networks, 	arXiv:1703.02161)

**Detailed Comments:**

Please see the weaknesses section.

**Justification Of Rating:**

Overall, the work seems to contribute to the progress of the area, considering that the contemporary methods also have similar performances. The direction of the methodology seems novel. Thus, while, there are some concerns about the approach, I believe that it can be shared with the community.

**Paper Type:**

methodological development

**Questions To Address In The Rebuttal:**

Most of the comments mentioned in the weaknesses, can be addressed in the Rebuttal

**Special Issue:**

no

---

> ### Author Response · Authors · 2020-03-28
> **Response to Reviewer 5**
>
> We thank you for the comments.  These comments were helpful in improving the final draft, and below we provide our response.
>
> Combining brain summaries with p-GCN:
> It’s a good suggestion. It could include interesting results but we made deliberate effort to leave out phenotypic data for the definition of population graphs.  The focus of our work was to eliminate the need of using phenotypic information because:
> (a) Using non-imaging data such as site information in creating an adjacency matrix reduces the transferability of the model to access new sites, as we would not know how to “connect” these patients to the graph.
> (b) The results from p- and s-GCN are comparable with s-GCN showing a marginal improvement at best. In the updated manuscript we will provide the results of a statistical test to quantify these observations. Also, we would like to iterate that s-GCN was a stepping stone towards our final model - the ss-GCN - which according to Table-2 has a much better performance.
> (c)  Anonymising structural scans are easier and most protective of privacy than using phenotypic data.
> It would be interesting to pursue this in a follow up paper.
>
> Variations in the results of ss-GCN:
> The large variations in the performance of ss-GCN across sites is a result of vast heterogeneity in datasets between sites. These occur either due to many reasons such as different SNR per site (a figure will be included in the appendix of the paper showing the different SNR) and different MRI image acquisition parameters at every site. Now, s-GCN has elements of the cross-correlation matrix as input compared to ss-GCN that have the brain summaries, thus the effect of these heterogeneities could be dramatically different.
>
> Rewording the sentence about sMRI and low-variance:
> Previous studies ( Su et al., 2012, Tyan et al., 2017 and Littmann et al., 2006) have shown associations between sMRI and age/sex/site. These studies also indicate, for example, that “brain-age” need not always coincide with the age reported. In order to avoid such uncertainties in establishing the edges on a graph, we resorted to the use of sMRI images directly —  giving us one variable to establish the relationship. We will reword the sentence you refer to, in order to clarify this point.
>
> Use of gamma:
> For defining graphs in p-GCN we used the exact same procedure as used in our baseline approach Parisot et al. (2018), where “Gamma” is defined differently depending on the type of phenotypic measure integrated in the graph. For categorical information such as subject’s sex, they define “Gamma” as the Kronecker delta function δ, meaning that the edge weight between subjects is increased if e.g. they have the same sex. Constructing edge weights from quantitative measures (e.g. subject’s age) is slightly less straightforward. In such cases, they defined “Gamma” as a unit-step function with respect to a threshold. Further details can be found in Section 2.2.2 of Parisot et al. (2018). Given the lack of clarity in our paper, we will add these details in the updated draft.
>
> Data for VAE pre-training:
> VAE was pre-trained on 5000 sMRI images from the UK Biobank database.  A link to download the trained VAE model will be revealed in the final draft of the paper.
>
> Comparison with other contemporary methods:
> The paper (arXiv:1703.02161) suggested by the reviewer uses metric learning on graphs. They cast the problem as a graph classification problem as opposed to node classification as we have done. Thus, comparing the methods would not be possible. The dataset used in that paper is a curated list from ABIDE-I and we have used a more heterogeneous sample that includes ABIDE-I and II. A discussion related to this will be added in the updated draft.
>
> References:
> [1] Longfei Su, Lubin Wang, and Dewen Hu. Predicting the age of healthy adults from structural mri by sparse
>  representation. In International Conference on Intelligent Science and Intelligent Data Engineering, pages 271–279. Springer, 2012.
> [2] Yeu-Sheng Tyan, Jan-Ray Liao, Chao-Yu Shen, Yu-Chieh Lin, and Jun-Cheng Weng. Gender differences in the structural connectome of the teenage brain revealed by generalized q-sampling mri. NeuroImage: Clinical, 15:376–382, 2017.
> [3] Arne Littmann, Jens Guehring, Christian Buechel, and Hans-Siegfried Stiehl. Acquisitionrelated morphological variability in structural mri. Academic radiology, 13(9):1055–1061, 2006.
> [4] Sarah Parisot, Sofia Ira Ktena, Enzo Ferrante, Matthew Lee, Ricardo Guerrero, Ben Glocker, and Daniel Rueckert. Disease prediction using graph convolutional networks: Application to autism spectrum disorder and alzheimer’s disease. Medical image analysis, 2018

---

### Official Review · AnonReviewer3 · 2020-03-20
**Interesting way of combining structural and functional data, but some weaknesses in experiments**

**Rating:** 3
**Confidence:** 4

**Summary:**

The authors propose a new approach for combining structural and functional MRI data in GCN analysis. The graph nodes represent subjects, where the nodes incorporate features derived from fMRI, while the edges describe the relationship between subjects are computed based on similarity of structural MRI. The method was tested on the full ABIDE I and II dataset and authors reported general improved performance using their approach over a recent GCN method that defines the graph edges based on population demographics.

**Strengths:**

1. Paper is fairly well written and easy to follow.
2. The proposed idea of combining structural and functional information through using one mode for edges and the other for node features appears novel, as usually functional information is used to define both nodes and edges, or edges are defined based on population or other nonimaging characteristics.  Structural information has been used to define graph edges but in the context of the graph representing the brain network for a single subject.
3. The authors present a fairly detailed comparison to a recent state-of-the-art approach that applied GCN in autism classification, including both intrasite 10-fold cross validation experiments as well as leave-one-site-out experiments.


**Weaknesses:**

1. Some important details are missing or confusing.
- It is not clear exactly what features are used to compute the edges for the competing p-GCN.
- The authors state the best atlas for p-GCN and s-GCN are used for the leave site out evaluation, and claim for s-GCN the best atlas was schaefer 400, but from Fig. 3 HO appears to be the best.

2. Experimental results are incomplete and some conclusions do not appear to be supported based on the presented info.
- The experiments only use one measure for classification performance - the accuracy. It would be better if other measures (eg sensitivity and specificity) were also shown and discussed.
- In Sec 4.1 in comparing p-GCN and s-GCN, the authors claim s-GCN performs better for 6 out of 9 atlases. Given the range of values, it seems to me that s-GCN may potentially truly do better in the first 5 out of 9 atlases. However, without some sort of significance testing, it is difficult to come to a conclusion, and it seems more like it is better half the time.
- In Sec 4.3 for the leave-one-site-out experiments, the authors claim that their approaches s-GCN and ss-GCN perform better than p-GCN in 4 out of 5 site with largest number of subjects. Again, it is hard to say that the numbers are better for some (is 61.1 significantly better than 60.9?). Also authors are only showing results for 5 out of the ~30 sites in the dataset. It would be interesting to compare the leave-one-out results across all these sites to get a better idea of whether the proposed approaches really improve classification performance.


**Justification Of Rating:**

The authors propose an interesting new method for defining the edges in a population graph, using the similarity in structural MRI between subjects. This method presents a new way of combining structural and functional data (structural data on edges, functional data on nodes of the graph). However, there are some weakness in the experimental section which make it unclear whether the proposed approach results in superior performance.

**Paper Type:**

both

**Questions To Address In The Rebuttal:**

Please see the points above.

Also, I am curious why the authors only tried one brain "summary" at a time - have they considered trying to include all the brain summaries together for the node features?

**Special Issue:**

no

---

> ### Author Response · Authors · 2020-03-28
> **Response to Reviewer 3**
>
> Thank you for the comments.  These comments were helpful in improving the final draft, and below we provide our response.
> Construction of population graph for p-GCN:
> The edges in p-GCN are created following Parisot et.al. 2018. (Section 2.2.2), where an edge is a function of phenotypic data such as sex, age and data acquisition site. For example, if two subjects are of the same sex, age-range or acquisition site, then an edge with a specific weight exists between them (Eq.1 in our paper). Details related to this aspect will be added in the updated draft.
>
> Label correction in Fig.  3:
> We thank the reviewer for spotting this typo. The label will be fixed in the final draft.
> More measures of classification performance: We have updated our manuscript to now include balance accuracy (the mean of sensitivity and specificity) and area under the ROC curve (AUC). These details are added in Section 4 of the updated draft.
>
> Significance testing for performance between s- and p-GCN:
> We will modify the statement to reflect the observation of the reviewer. The results from p- and s-GCN are comparable with s-GCN showing a marginal improvement at best. In the updated manuscript we will provide the results of a statistical test to quantify these observations. Also, we would like to iterate that s-GCN was a stepping stone towards our final model - the ss-GCN - which according to Table-2 has a much better performance. We will add a discussion related to the analysis of edges for the population graphs of p-GCN and s-GCN based on  (i) Overall degree distribution, (ii) Degree distribution split according to class.  For a better measure of classification performance, we now also report balance accuracy score and area under the ROC curve (AUC). These details are added in Section 4 of the updated draft.
> Note: The performance of s-GCN is lower mostly in the atlases we called JAMA-x in Fig.3. We would like to highlight that this is not a “true” atlas like the rest of them and is composed of ICA components derived from a previous study on Autism [Cerliani et al., JAMA Psychiatry, 2015]. These are spatially distributed regions and the number of components or regions is rather low [7-52]. It is becoming prudent in the community to give lesser importance to these “atlases”. We will add this remark in the updated draft.
>
> Leave-one-site-out for all sites:
> We decided to use only the top-5 sites in numerosity of subjects to perform the leave-one-site cross validation in order to get a large (>100) sample in our test set (Ref. Table in Appendix B). Sites with just a few samples wouldn’t yield a very reliable estimate of the performance. Secondly, performing cross validation across >30 sites (i.e, 30 splits) would have been computationally challenging.
>
> Combining brain summaries:
> Based on your suggestion, we first performed an experiment where we explicitly combined the classifiers from all brain summaries by making a voting classifier per fold. The result did seem to improve with a mean accuracy of 64.23% and AUC score of 68.31%. We will add a detailed discussion of this experiment in our updated draft.
> As per as combining them implicitly as input, we used a 3D CNN with brain summaries as channels to access the performance of using all summaries simultaneously. Unfortunately, this experiment did not yield good results compared to many of the individual summaries. Also, performing a full cross validation using all brain summaries as channels would have been computationally intensive. Thus, we omitted the experiment from the scope of this paper.
>
> References:
> [1] Cerliani, L., Mennes, M., Thomas, R. M., Di Martino, A., Thioux, M., & Keysers, C. (2015). Increased functional connectivity between subcortical and cortical resting-state networks in autism spectrum disorder. JAMA psychiatry, 72(8), 767-777
> [2] Sarah Parisot, Sofia Ira Ktena, Enzo Ferrante, Matthew Lee, Ricardo Guerrero, Ben Glocker, and Daniel Rueckert. Disease prediction using graph convolutional networks: Application to autism spectrum disorder and alzheimer’s disease. Medical image analysis, 2018

---

### Meta-Review · Area_Chair1 · 2020-04-06
**MetaReview of Paper183 by AreaChair1**

**Rating:** 3
**Recommendation For Accepted Papers:** Poster

**Metareview:**

3 out of 4 reviewers recommended weak acceptance of this work, while 1 reviewer recommended strong acceptance. After reading their comments and the corresponding answers given by the authors in their rebuttal, I think this work should be accepted for publication at MIDL.

Please, when submitting the Camera Ready version, take into account the suggestions made by the reviewers which will improve the quality of your final submission.

**Paper Type:**

both

**Special Issue:**

no

---

### Decision · Program_Chairs · 2020-04-11

Accept